# DISTRIBUTED TRAINING ACROSS THE WORLD

## ABSTRACT

Traditional synchronous distributed training is performed inside a cluster since it requires high bandwidth and low latency network (*e.g.* 25Gb Ethernet or Infiniband). However, in many application scenarios, training data are often distributed *across many geographic locations*, where physical distance is long and latency is high. Traditional synchronous distributed training cannot scale well under such limited network conditions. In this work, we aim to **scale distributed learning under high-latency network**. To achieve this, we propose **D**elayed and **T**emporally **S**parse (DTS) update that enables synchronous training to tolerate extreme network conditions without compromising accuracy. We benchmark our algorithms on servers deployed across three continents in the world: London (Europe), Tokyo (Asia), Oregon (North America) and Ohio (North America). Under such challenging settings, DTS achieves $90\times$ speedup over traditional methods without loss of accuracy on ImageNet.

## 1 INTRODUCTION

Deep neural networks have demonstrated much success in solving large-scale machine learning problems (Krizhevsky et al., 2012; Simonyan & Zisserman, 2014; He et al., 2016). However, training deep neural networks may take days or even weeks to converge. In order to enable training in a reasonable time, distributed training is an important technique and gains increasing attention (Li et al., 2014; Dean et al., 2012; Recht et al., 2011; Goyal et al., 2017; Jia et al., 2018). To maintain a good scalability, a *low latency* and *high bandwidth* network is essential for most modern distributed systems. Existing frameworks (Chen et al., 2015; Xing et al., 2015; Moritz et al., 2015; Abadi et al., 2015; Akiba et al., 2017; Paszke et al., 2017; Sergeev & Del Balso, 2018) all require high-end network infrastructure such as 25Gbps Ethernet or Infiniband where bandwidth is as large as 10 to 100 Gbps and latency is as small as 1 us.

Bandwidth is easy to increase (*e.g.* stacking hardware) but latency is hard to improve (physical limits). For example, if we have two servers located at Shanghai and Boston respectively, even at the speed of light and direct air distance, it still takes 78ms[*] to send and receive a packet. In real world scenario, the latency can be only worse (around 700ms) because indirect routing between internet service providers (ISP) and queuing delay in switches. Such high latency cause severe scalability[†] issue for distributed training. As shown in Fig. 1, traditional distributed training algorithm scales poorly under such large latency.

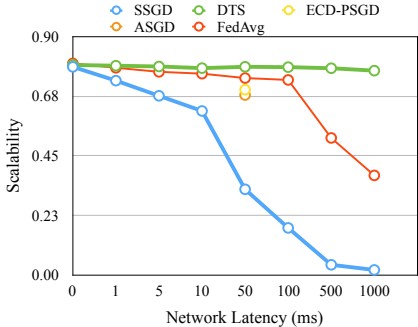

Figure 1: DTS maintains a good scalability when network latency increases, while the performance of conventional algorithms degrades quickly.

In many scenarios, the training data involves privacy-sensitive information, such as personal medical history (Jochems et al., 2016; 2017) and keyboard input history (McMahan et al., 2016; Konen et al., 2016; Bonawitz et al., 2019), thus cannot be centralized to a data center due to security and legacy concerns (GDPR, 2016). When datasets are distributed across many

---

[*]$11,725\text{km} \times 2/(3 \times 10^8 \text{m/s}) = 78.16\text{ms}$. Information collected from Google Maps.
[†]If a system can achieve $M$ times speedup on $N$ machines, the scalability is defined as $M/N$.

Figure 2: Network conditions across different continents (Left) and the comparison between connections inside a cluster (Right). Different from training inside a data center, long-distance distributed training suffers from high latency, which proposes a severe challenge to scale across the world.

locations across public clouds, private clusters and even edge devices, it is impossible to setup a low latency network under long-distance connections and thereby hurts the scalability of training.

In this work, we enable **scalable distributed training across different geographical locations** with long distance and high latency network connection. We propose *Delayed Update* to tolerate latency by putting off the synchronization barrier to a later iteration; we also propose *Temporally Sparse Update* to amortize the latency and alleviate congestion by reducing the synchronization frequency. To ensure no loss of accuracy, we design a novel error compensation to overcome the staleness for both vanilla and momentum SGD. We focus on the widely adopted synchronous update using data parallelism. As shown in Fig. 1, DTS can maintain high scalability even when latency is as high as 1000ms. This result is also better than existing state of art technologies such as ECD-PSGD (results copied directly from their original paper (Tang et al., 2018a))

We benchmark our DTS under a challenging settings: training a deep neural network on four AWS P3 instances located in different continents of the world (Fig. 2). The measured latency is as large as 277ms (compared to internal latency 2us). In this case, the naive distributed synchronous SGD (SSGD) can only achieve a poor scalability of 0.008. It means in this setting, distributed training with 100 servers is even slower than single machine (0.8 v.s. 1.0). Meanwhile DTS achieves scalability of 0.72 without compromising the accuracy. In conclusion, our contributions are listed below:

- We propose delayed update to tolerate the latency and temporally sparse update to amortize the latency. While preserving the accuracy, delayed update tolerates up to 6 seconds latency and temporally sparse update reduced the traffic congestion by $20\times$.

- We theoretically justify the convergence rate of our proposed algorithms. We show that both algorithms can be as fast original SGD while scaling well under high latency.

- With servers and data distributed in four different countries across the world, we can train ResNet-50 on ImageNet with scalability. To our best knowledge, DTS is the first work that can achieve scalable synchronous distributed training under such high latency.

## 2 RELATED WORK

**Distributed Learning** becomes ever more important as the sizes of both datasets and models increase. Many studies have been made to explore the efficiency, both at the algorithm level (Li et al., 2014; Dean et al., 2012; Recht et al., 2011; Goyal et al., 2017; Jia et al., 2018) and at the framework level (Chen et al., 2015; Akiba et al., 2017; Abadi et al., 2015; Paszke et al., 2017; Sergeev & Del Balso, 2018). In most of distributed algorithms, each node performs computation and exchange updates through network. A key component to improve scalability is to reduce the communication-to-computation ratio. The communication cost is determined by latency and bandwidth. Conventional studies focus on reducing bandwidth requirements as the latency of internal traffic is usually low.

**Gradient quantization / compression** has been proposed to reduce the data to be transferred. One-bit gradient gradient (Seide et al., 2014) achieves $10\times$ speedup using 20 GPUs on text-to-speech task. QSGD (Alistarh et al., 2016) and Terngrad (Wen et al., 2017) further improves the trade-off between accuracy and gradient precision. But the theoretical limit of the quantization cannot go beyond 32. To overcome the limitation, gradient sparsification using predefined static threshold (Strom, 2015; Aji &

Heafield, 2017) and dynamic compression ratio (Chen et al., 2018) demonstrate that 99% gradients can be pruned with negligible degradation on model performance. (Sattler et al., 2018) combines both quantization and compression to push the ratio to a new level. DGC (Lin et al., 2017) and DoubleSequeeze (Tang et al., 2019) explore how to compensate the error to preserve the accuracy better. The convergence of quantization and compression is also discussed in (Tang et al., 2019; Jiang & Agrawal, 2018; Alistarh et al., 2018). However, latency remains an issue, especially when servers are not in the same physical location.

**Learning on decentralized data** has become increasingly popular recently as the growing awareness of data privacy (GDPR, 2016). For example, Federated Learning (Google, 2017) aim to jointly train a model without centralizing the data and have been used to train models for medical treatments across multiple hospitals (Jochems et al., 2016), analyze patient survival situations from various countries (Jochems et al., 2017) and build predictive keyboards to improve typing experience (McMahan et al., 2016; Google, 2017; Bonawitz et al., 2019). However, existing federated learning works do not scale well under high latency network. An orthogonal exploration is Decentralized Training where only partial synchronization is performed in each update, such as AD-PSGD (Lian et al., 2017) and $D^2$ (Tang et al., 2018b). None of them has been evaluated on large learning tasks yet.

**Asynchronous SGD** (ASGD) is derived from (Tsitsiklis et al., 1986) and has advantages in unstable latency and fault tolerance. Different from synchronous SGD (SSGD), ASGD relaxes synchronization by allowing training on inconsistent models and powers many successful applications such as HOGWILD! (Recht et al., 2011), BUCKWILD! (De Sa et al., 2015) and dist-belief (Dean et al., 2012). However, most of them are implemented through *parameter server* (Dean et al., 2012; Chilimbi et al., 2014; Li et al., 2014) and leads to problems like communication congestion and resource congestion when scaling up. Moreover, training on inconsistent models leads to different behaviors compared to training on a single device. Though there are studies discussing the convergences (Recht et al., 2011; De Sa et al., 2015) and showing that ASGD converges as good as SSGD (Lian et al., 2015), synchronous SGD (SSGD) is usually preferred in practice (Goyal et al., 2017; Sun et al., 2019) because it has consistent behaviors when increasing the number of machines and therefore is easy to develop and deploy.

## 3 APPROACH

To motivate this section, we first review the mechanism of synchronous distributed training: at each step, each node will first compute gradients locally, then they wait for the collective operation to transmit gradients to each other to calculate the average. They use the averaged gradient to update the weight and continue to the next step. As shown in right of Fig. 3a, when the communication increases , the whole training process would be drastically slowed down.

To address the problem, our proposed algorithm contains two parts: *delayed update* (Fig. 3b) and *temporally sparse update* (Fig. 3c). With the delayed update, instead of waiting for average gradients to come back, it puts off the synchronization barrier to steps later to tolerate the latency. With the temporally sparse update, the synchronization frequency is reduced to amortize latency.

Throughout the paper, we use the following notions:

- $\gamma$: learning rate.
- $t$: iteration steps that the synchronization barrier is delayed.
- $p$: iteration steps between temporally sparse update intervals.
- $u_{(i,j)}$: momentum at iteration $i$ on worker $j$.
- $u'_{(i,j)}$: error compensated momentum at iteration $i$ on worker $j$.

- $w_{(i,j)}$: model weights at iteration $i$ on worker $j$.
- $w'_{(i,j)}$: error compensated weights at iteration $i$ on worker $j$.
- $\nabla w_{(i,j)}$: local gradients at iteration $i$ on worker $j$.
- $\overline{\nabla w_{(i)}}$: global averaged gradients at iteration $i$.

### 3.1 DELAYED UPDATE

In modern implementations of vanilla SSGD, the transmission is partially pipelined with back-propogation: Synchronizing gradients of $n^{\text{th}}$ layer can be performed simultaneously with back-propagating $(n-1)^{\text{th}}$ layer. However, it is not enough to cover communication since latency on

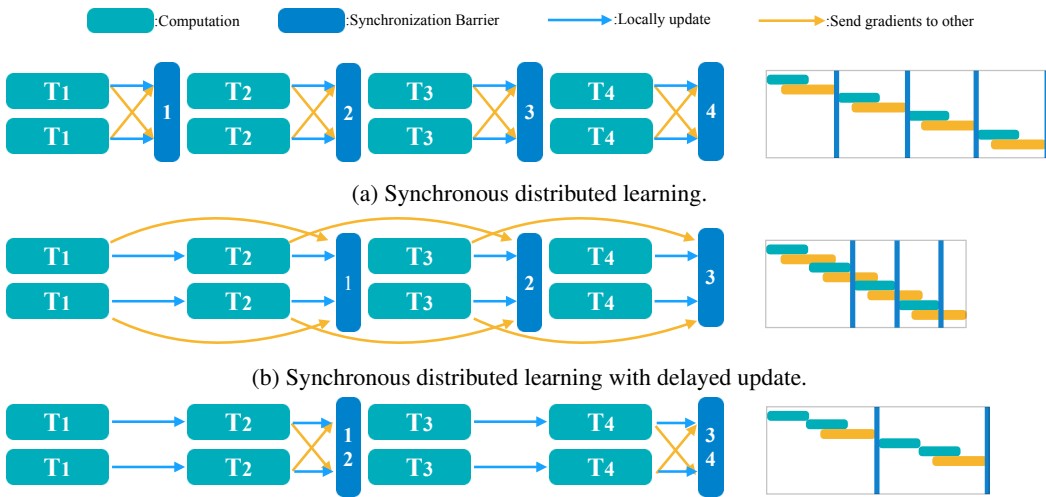

(a) Synchronous distributed learning.

(b) Synchronous distributed learning with delayed update.

(c) Synchronous distributed learning with temporally sparse update.

Figure 3: The visualization of our proposed techniques (*Left*) and the execution pipeline (*Right*). Yellow lines and blocks indicate communication, while green ones indicate computation. Delayed update covers communication by computation and requires the same number of synchronization. Temporally sparse reduces the synchronization frequency and improve the amortized latency.

long-distance connection is usually larger ($\geq$ 100ms) than propagating a layer ($\approx$ 11ms). To relax the limitation, we design delayed update to put off the the synchronization barrier by $n$ steps as shown in Fig. 3b. To ensure minimal loss on accuracy, we modify the gradient update formula with error compensations.

To understand, we start with vanilla SGD: Consider a worker with index $j$, the weights at timestamp $n$ during distributed training are derived via

$$w_{(n,j)} = w_{(n-1,j)} - \gamma \overline{\nabla w_{(n-1)}} = w_{(0,j)} - \gamma \sum_{i=0}^{n-1} \overline{\nabla w_{(i)}} \tag{1}$$

where $\overline{\nabla w_{(n-1)}}$ is the averaged gradients calculated by gathering all local gradients $\nabla w_{(n-1,j)}$. In SSGD, the synchronization has to finish before $(n+1)^{\text{th}}$ forwarding. In delayed update, we *delay* the barrier to $(n+t)^{\text{th}}$ iteration and because of the delay, $\overline{\nabla w_{(n-1)}}$ is not received at timestamp $n$. We use local gradients to update instead.

$$w_{(n,j)} = w_{(n-1,j)} - \gamma \nabla w_{(n-1,j)} \tag{2}$$

Meanwhile, stale gradients $\overline{\nabla w_{(n-t)}}$ arrive. At timestamp $n - t$, we performed the gradient step using local gradients $\nabla w_{(n-t,j)}$, which should be the global averaged $\overline{\nabla w_{(n-t)}}$ in SSGD. We correct the mismatch caused by replacing $\nabla w_{(n-t,j)}$ with $\overline{\nabla w_{(n-t)}}$. From Eq. 1, it adds a compensation term, which is the difference of two gradients.

$$w'_{(n,j)} = w_{(n-1,j)} - \gamma \underbrace{\nabla w_{(n-1,j)}}_{\text{local gradients}} - \gamma (\underbrace{\overline{\nabla w_{(n-t)}} - \nabla w_{(n-t,j)}}_{\text{error compensation}})$$
$$= w_{(n,j)} - \gamma (\overline{\nabla w_{(n-t)}} - \nabla w_{(n-t,j)}) \tag{3}$$

Next, we consider momentum SGD. With momentum $m$, the weight at timestamp $n$ is updated by

$$u_{(n,j)} = m u_{(n-1,j)} + \nabla w_{(n,j)} \quad w_{(n,j)} = w_{(n-1,j)} - \gamma u_{(n-1,j)} \tag{4}$$

We rewrite the cumulative form to general form

$$u_{(n,j)} = \sum_{i=0}^{n} m^i \nabla w_{(n-i,j)}$$
$$= \sum_{i=0}^{n} m^{n-i} \nabla w_{(i,j)}$$

(5)

$$w_{(n,j)} = w_{(0,j)} - \gamma \sum_{i=0}^{n-1} u_{(i,j)}$$
$$= w_{(0,j)} - \gamma \sum_{i=0}^{n-1} \sum_{k=0}^{i} m^{i-k} \nabla w_{(k,j)}$$
$$= w_{(0,j)} - \gamma \sum_{i=0}^{n-1} \sum_{k=0}^{n-1-i} m^k \nabla w_{(i,j)}$$

(6)

Then, we can derive the compensated momentum

$$u'_{(n,j)} = m u_{(n-1,j)} + \nabla w_{(n-1,j)} + m^t (\overline{\nabla w_{(n-t)}} - \nabla w_{(n-t,j)})$$
$$= u_{(n,j)} + m^t (\overline{\nabla w_{(n-t)}} - \nabla w_{(n-t,j)})$$

(7)

and weights similarly

$$w'_{(n,j)} = w_{(n-1,j)} - \gamma u_{(n-1,j)} - \gamma \sum_{k=0}^{t-1} m^k (\overline{\nabla w_{(n-t)}} - \nabla w_{(n-t,j)})$$
$$= w_{(n,j)} - \gamma \sum_{k=0}^{t-1} m^k (\overline{\nabla w_{(n-t)}} - \nabla w_{(n-t,j)})$$

(8)

In original synchronous stochastic gradient descent (SSGD), every worker receives averaged gradients from $n-1$ at timestamp $n$. This requires collective operation (*e.g.* Allreduce) to execute quickly; otherwise, it will slow down the training process. In delayed update, the barrier of the collective operation is moved from timestamp $n$ to $n+t-1$. It means as long as the gradients are transmitted within $t \times T_{\text{compute}}$, the training is not blocked. To put some real numbers, assume we are training ResNet-50 on Nvidia Tesla V100, each step takes around 300ms. If we delay the update by 20 steps, even with latency as large as 6 seconds, the communication still can be fully covered by computations. We attach the proof of convergence in Appendix A.2.1 and show this is as fast as SSGD.

## 3.2 TEMPORALLY SPARSE UPDATE

Delayed update provides a good solution to tolerate latency. However, there still remains two issues and network congestion is the first one. In the original SSGD, only one transmission is performed during one iteration. But when using delayed update, there can be up to $t$ transmissions at the same time. Without a special hacking to low-level network driver, gradients sent earlier may arrive later, which is not what we want. The second one is latency variance. In real world network, especially for those long-distance connections, the latency varies in a large range. Even though the average latency is small, some accidentally high latency can slow the system.

To address, we adapt temporally sparse update to reduce the communication frequency and design corresponding error compensation to guarantee the accuracy. As demonstrated in Fig. 3c, temporally sparse update only synchronizes for every $p$ steps. For those steps where synchronization is not performed, each worker updates its model with local gradients. For those steps where synchronization is performed, each worker calculates error compensations based averaged information. As a result, the networking congestion is alleviated and the networking latency is amortized across multiple local update steps.

For vanilla SGD, like the delayed update, we can derive the compensation term for temporally sparse update from Eq. 1, which is the difference between accumulated gradients. Note the compensation in temporally sparse update only applies when $n \pmod p \equiv 0$.

$$w'_{(n,j)} = w_{(n,j)} - \gamma \left( \sum_{i=n-p}^{n-1} \overline{\nabla w_{(i)}} - \sum_{i=n-p}^{n-1} \nabla w_{(i,j)} \right)$$

(9)

| | Bandwidth | Non-computing time |
|---|---|---|
| Synchronous SGD | $2\|w\|/T_{\text{training}}$ | $T_{\text{communicate}} - T_{\text{overlap}}$ |
| Delayed Update (t) | $2\|w\|/T_{\text{training}}$ | $max(0, T_{\text{communicate}} - T_{\text{overlap}} - t \times T_{\text{compute}})$ |
| Temporal Sparsity (p) | $4\|w\|/(p \times T_{\text{training}})$ | $(T_{\text{communicate}} - T_{\text{overlap}})/p$ |
| Delayed Update (t) + Temporal Sparsity (p) | $4\|w\|/(p \times T_{\text{training}})$ | $max(0, (T_{\text{communicate}} - T_{\text{overlap}} - t \times T_{\text{compute}}))/p$ |

Table 1: The networking requirements for different training strategies.

For SGD with momentum acceleration, based on accumulated gradients is not enough to compensate both $\nabla w_{(n,j)}$ and $u_{(n,j)}$. We need to transmit an extra term to correct the momentum:

$$u'_{(n,j)} = u_{(n,j)} + \left( \sum_{i=n-p}^{n-1} m^{n-i}\overline{\nabla w_{(i)}} - \sum_{i=n-p}^{n-1} m^{n-i}\nabla w_{(i,j)} \right)$$

$$w'_{(n,j)} = w_{(n,j)} - \gamma \left( \sum_{i=n-p}^{n-1} \sum_{k=0}^{n-1-i} m^k \overline{\nabla w_{(i)}} - \sum_{i=n-p}^{n-1} \sum_{k=0}^{n-1-i} m^k \nabla w_{(i,j)} \right) \tag{10}$$

$$= w_{(n,j)} - \gamma \left( \sum_{i=n-p}^{n-1} \sum_{k=n-p}^{i} m^{i-k}\overline{\nabla w_{(k)}} - \sum_{i=n-p}^{n-1} \sum_{k=n-p}^{i} m^{i-k}\nabla w_{(k,j)} \right)$$

Let $S_{(i,j)} = \sum_{k=n-p}^{i} m^{i-k}\nabla w_{(k,j)}$. Note $S_{(i+1,j)} = mS_{(i,j)} + \nabla w_{(i+1,j)}$ and $S_{(n-p+1,j)} = \nabla w_{(n-p,j)}$, the formula can be written as:

$$u'_{(n,j)} = u_{(n,j)} + (\overline{S_{(n)}} - S_{(n,j)})$$

$$w'_{(n,j)} = w_{(n,j)} - \gamma \left( \sum_{i=n-p}^{n-1} \overline{S_{(i)}} - \sum_{i=n-p}^{n-1} S_{(i,j)} \right) \tag{11}$$

Note temporally sparse update is different from gradient accumulation (forward and backward $N$ times, accumulate the gradients and optimizer steps only 1 time). Gradient accumulation is equivalent to increasing the batch size and requires to adjust learning rate and the parameters of batch normalization to ensure smooth convergence. Temporally sparsity updates model every iteration and compensate error every $N$ steps. It does not require any changes to the hyper-parameters. We prove the convergence of temporally sparse update in Appendix A.2.2 and demonstrate that it enjoys the same convergence rate as SSGD.

### 3.3 DELAYED AND TEMPORALLY SPARSE UPDATE

The delayed technique and temporally sparse update can be combined together. The compensation formulas for vanilla SGD is

$$w'_{(n,j)} = w_{(n,j)} - \gamma \left( \sum_{i=n-t-p}^{n-t-1} \overline{\nabla w_{(i)}} - \sum_{i=n-t-p}^{n-t-1} \nabla w_{(i,j)} \right) \tag{12}$$

and for momentum SGD is

$$u'_{(n,j)} = u_{(n,j)} + m^t(\overline{S_{n-t}} - S_{(n-t,j)})$$

$$w'_{(n,j)} = w_{(n,j)} - \gamma m^t \left( \sum_{i=n-t-p}^{n-t-1} \overline{S_i} - \sum_{i=n-t-p}^{n-t-1} S_{(i,j)} \right) \tag{13}$$

Original ring allreduce transfers $2\|w\|$ packets (Thakur et al., 2005). Though temporally sparse update doubles data transferred, the temporal sparsity $t$ is always larger than 2. As a consequence, combing two methods can further improve scalability under challenging networking ( Tab. 1).

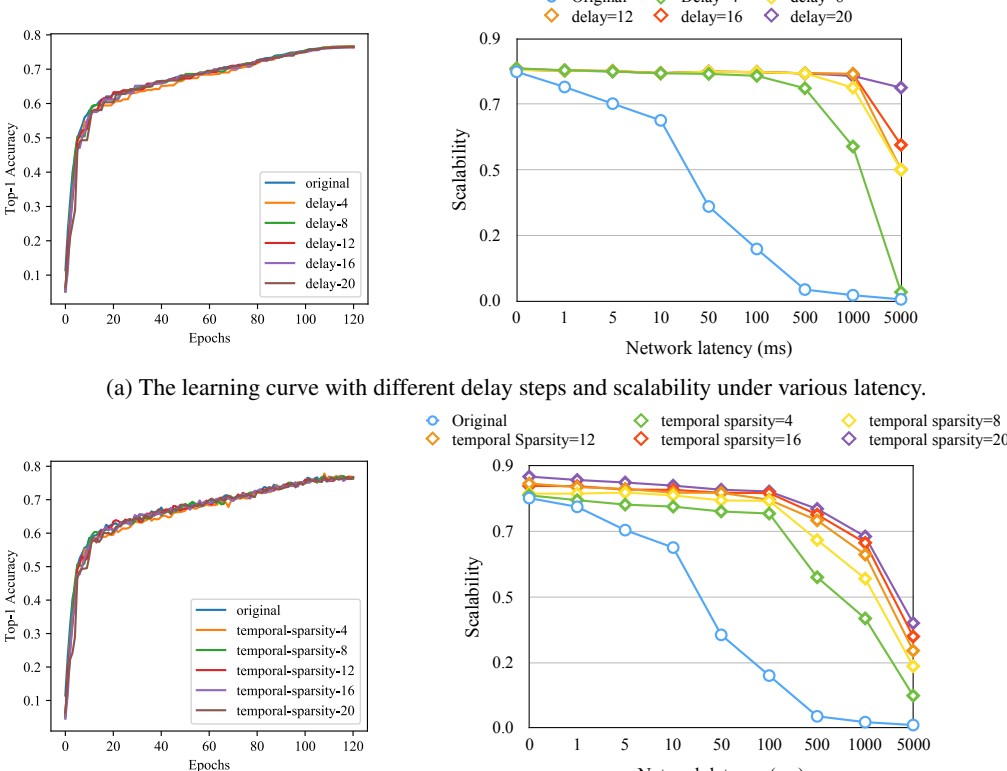

(a) The learning curve with different delay steps and scalability under various latency.

(b) The learning curve with different temporal sparsity and scalability under various latency.

Figure 4: Compare the top-1 accuracy and scalability on ImageNet training.

## 4 EXPERIMENT

### 4.1 EXPERIMENT SETTINGS

We evaluate our DTS on image classification task. We studies ResNet-50 (He et al., 2016) on ImageNet (Deng et al., 2009). ImageNet contains 1.2 million training images and 50,000 validation images in 1000 classes. We train ImageNet model with momentum SGD using warmup strategy (Goyal et al., 2017) and cosine anneal learning rate decay (Loshchilov & Hutter, 2016) with 120 epochs. Standard data augmentations random crop and flip are adapted. We set the initial learning rate 0.0125 for which grows linearly w.r.t number of GPUs. For a fair comparison, the training settings are set the same among all experiments.

When performing ablation studies under different latency, we synthesis the network condition using *netem*[‡]. When training across the world, the limited network is benchmarked using *qperf*[§]. We implement DST in PyTorch framework (Paszke et al., 2017) and adapt Horovod (Sergeev & Del Balso, 2018) as the distributed training backend. To make results reproducible, we conduct our experiment on 16 standard Amazon *p.large* EC2 instances with eight Nvidia Tesla V100 on each and will release the codebase when less anonymous.

### 4.2 RESULTS OF DELAYED UPDATE

Fig. 4a shows the performance and scalability for delayed update with different latency. We start from delay steps 4 and gradually increase it. The learning curve shows that with our proposed error compensation delayed update demonstrates robust convergence and comparable performance compared vanilla SSGD, even when delay interval is as large as 20.

---

[‡] https://wiki.linuxfoundation.org/networking/netem
[§] https://linux.die.net/man/1/qperf

|              | Top-1% |
| ------------ | ------ |
| Original SGD | 76.63  |
| D=4, T=4, C=1% | 76.15 |
| D=8, T=8, C=1% | 76.32 |
| D=12, T=8, C=1% | 76.18 |
| D=20, T=12, C=1% | 75.81 |

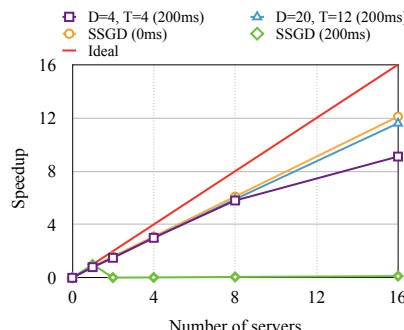

(a) Delayed updates preserves the accuracy while tolerate a large latency. D, T, C indicate delayed update, temporal sparsity and gradient compression ratio respectively. The results are evaluated on ImageNet dataset.

(b) Better speedup and scalability with DTS. Measured on servers across the world.

Table 2: DTS demonstrates good scalability and accuracy on world-wide high latency training.

In original SSGD and modern deep learning frameworks, though the transmission is partially pipelined with backpropogation: Synchronizing gradients of $n^{th}$ layer can be performed simultaneously with back-propagating $(n-1)^{th}$ layer, it is not enough to fully cover communication since latency on long distance connection is usually larger ( $\geq$ 100ms) than propagating a layer (11ms). By delaying the synchronization, delayed update provides much more spare time for transmission and thus scales better. As shown in Fig. 4a, the larger the delayed interval is, the better tolerance towards latency. The scalability keeps stable until maximum tolerance is exceeded.

### 4.3 RESULTS OF TEMPORALLY SPARSE UPDATE

The connection of real-world network has many bothering issues such as unstable latency and congestion on routers. Temporally sparse is introduced to amortize these via reducing the synchronization frequency. The results are shown in Fig. 4b(*Left*): we start with temporal sparsity 4 and increase to 20. With proposed compensation, the temporally sparse update has a negligible loss on accuracy. Fig. 4(*Right*) exhibits how temporally sparse training scales under different network latency. The scalability degrades slower than because temporally sparse update amortizes the latency.

### 4.4 DISTRIBUTED TRAINING ACROSS THE WORLD

With two powerful techniques, we are close to our original target of distributed training across world. To evaluate DTS, we deploy located at four different locations across the world: Tokyo(Japan), London (U.K), Oregon (U.S.A) and Ohio(U.S.A). The latency information is shown in Fig.2. The real latency across the ring allreduce is about 479ms through Ethernet. On V100 GPU, ResNet-50 takes around 300ms to forward and backward. Therefore delayed update with steps more than 4 is already enough to tolerate such a network. In terms of latency variance and network congestion, we integrate temporally sparse update to alleviate which drastically reduces the bandwidth requirement by $t$ times. However, there is still a bottleneck on bandwidth for cross-continent connections (*e.g.* Tokyo to London). Though theoretically bandwidth is achievable by switching to better internet service providers, AWS does not provide such an option. As an economical alternative, we adapt deep gradient compression (Lin et al., 2017) to work through. As shown in Tab. 2a, DTS has little effect on training accuracy. Meanwhile, DTS demonstrates strong scalability (0.72) under high latency (200ms between workers) and this performance is close to what conventional algorithms achieved inside a data center (speedup ratio 0.78 and 1us latency).

## 5 CONCLUSION

In this paper, we propose two novel delayed update and temporally sparse update methods to scale synchronous distributed training on servers located at different continents across the world. We demonstrate that delayed synchronization and temporally sparse update are both accuracy preserving while tolerate poor network conditions. We believe that our work will open future avenues for a wide range of decentralized learning applications.

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

# A  APPENDIX

## A.1  CONNECTION TO VARIANCE REDUCTION APPROACH

A lot of recent work has been devoted to the study of variance reduction techniques in convex optimization, including (Johnson & Zhang, 2013; Shalev-Shwartz & Zhang, 2013b; Xiao & Zhang, 2014; Schmidt et al., 2017; Defazio et al., 2014; Mairal, 2015). The vanilla SVRG algorithm has the update

$$w_n = w_{n-1} - \eta \left[ \nabla f_i(w_{n-1}) + \overline{\nabla f(\tilde{w})} - \nabla f_i(\tilde{w}) \right], \tag{14}$$

where $\tilde{w}$ is a snapshot reference point updated every constant iteration. These methods are able to achieve linear convergence rates for smooth strongly-convex optimization problems, which significantly improve the sub-linear rate of vanilla SGD. Extension to multiple setting are investigated, including acceleration (Shalev-Shwartz & Zhang, 2013a; Lin et al., 2015; Allen-Zhu, 2017), streaming (Frostig et al., 2015; Lei & Jordan, 2016), synchronous distributed setting (Lee et al., 2015), data augmentation (Bietti & Mairal, 2017). Extension of variance reduction approaches to general non-convex problems has also been studied in (Allen-Zhu & Hazan, 2016; Reddi et al., 2016). Despite these theoretical progress, variance reduction technique does not appear to be helpful for training deep neural network (Defazio & Bottou, 2018).

Our proposed method could be viewed as a delayed variant of SVRG in the decentralized setting. The update (15) in our method mimic the update of vanilla SVRG (14), with the difference that the average gradient is obtained across machine base on local parameters. This variance reduction technique empirically stabilized the asynchronous gradients and handle the staleness, which is able to achieve state-of-the-art performance in modern deep learning setting. Moreover, we obtain close linear speed up with respect to the number of machine, which is the essential goal of distributed optimization.

## A.2  THEORETICAL JUSTIFICATION

Our proof closely follows the analysis in (Yu et al., 2019). We consider the following non-convex optimization problem

$$\min_w f(w) = \frac{1}{J} \sum_{j=1}^{J} f_j(w),$$

where $J$ denotes the number of workers. Moreover, each $f_j = \mathbb{E}_{\zeta_j}[F_j(w, \zeta_j)]$. In the empirical minimization setting, $\zeta_j$ corresponds to a mini-batch sampling strategy and $f_j$ could again be expressed as a finite sum $f_j = \frac{1}{K_j} \sum f_{k_j}$. Throughout this paper, we assume the optimization satisfies the following assumptions.

**Assumption 1** (*L*-smooth). *Each function $f_j(x)$ is L-smooth, i.e. for all $x, y \in \mathbb{R}^d$ and $i \in [1, J]$,*

$$||\nabla f_j(x) - \nabla f_j(y)|| \leq L||x - y||.$$

**Assumption 2** (Bounded gradients and variances). *There exists a positive constant $G$ and $\sigma$ such that*

$$\mathbb{E}_{\zeta_j}||\nabla F_j(w; \zeta_i)||^2 \leq G^2, \forall w, \forall j$$
$$\mathbb{E}_{\zeta_j}||\nabla F_j(w; \zeta_j) - \nabla f_j(w)||^2 \leq \sigma^2, \forall w, \forall j$$

We prove that our algorithms have the convergence rate

$$O(\frac{1}{\sqrt{JN}}) + O(\frac{c^2 J}{N})$$

where $J$ is the number of workers, $N$ is iterations and $c$ is staleness in our algorithm ($t$ for delayed steps and $d$ for temporal sparsity). With proper settings, our algorithms are as fast as original SSGD $O(1/\sqrt{JN})$. The detailed proof follows below.

### A.2.1 PROOF OF THE CONVERGENCE OF DELAYED UPDATE ALGORITHM

---

**Delayed Update Algorithm**

1. Initialize $w_0$ and initialize each worker with $w_{0,j} = w_0$ for $j \in [1, J]$.

2. **For** iteration $n = 0, 1, \cdots$

   On each local worker $j$, sample a stochastic gradient $\nabla w_{(n-1,j)}$ and update:
   ○ If $n < t$, update with standard SGD rule:

   $$w_{(n,j)} = w_{(n-1,j)} - \gamma \nabla w_{(n-1,j)}$$

   ○ Else:

   $$w_{(n,j)} = w_{(n-1,j)} - \gamma(\nabla w_{(n-1,j)} + \overline{\nabla w_{n-t}} - \nabla w_{(n-t,j)}) \qquad (15)$$

   where

   $$\overline{\nabla w_{n-t}} = \frac{1}{J} \sum_{j=1}^{J} \nabla w_{(n-t,j)}$$

3. **Output** $\overline{\nabla w_{(n)}} = \frac{1}{J} \sum_{j=1}^{J} w_{(n,j)}$

---

Notably, the delayed update formula can rewritten as

$$w_{(n,j)} = \underbrace{w_0 - \gamma \sum_{i=0}^{n-t} \overline{\nabla w_i}}_{\text{invariant of } j} - \gamma \sum_{i=n-t+1}^{n-1} \nabla w_{(i,j)}, \qquad \text{(Delayed)}$$

where the first quantity is commonly shared among all the workers and the second term corresponds to local updates.

**Lemma A.1.** *Let $(w_{(i,j)})$ be the sequence generated on the $j$-th worker according to Delayed Update Algorithm, then for any $j_0 \in [1, J]$*

$$\mathbb{E}\left[\|\overline{\nabla w_{(n)}} - w_{(n,j_0)}\|^2\right] \le 4\gamma^2 t^2 G^2$$

*Proof.* From the delayed update formula Delayed, we have,

$$\mathbb{E}\left[\|\overline{\nabla w_{(n)}} - w_{(n,j_0)}\|^2\right]$$

$$= \gamma^2 \mathbb{E}\left[\left\|\frac{1}{J}\sum_{j=1}^{J}\sum_{i=n-t+1}^{n-1}\nabla w_{(i,j)} - \sum_{i=n-t+1}^{n-1}\nabla w_{(i,j_0)}\right\|^2\right]$$

$$\le 2\gamma^2 \mathbb{E}\left[\left\|\frac{1}{J}\sum_{j=1}^{J}\sum_{i=n-t+1}^{n-1}\nabla w_{(i,j)}\right\|^2\right] + \mathbb{E}\left[\|\sum_{i=n-t+1}^{n-1}\nabla w_{(i,j_0)}\|^2\right]$$

$$\le \frac{2\gamma^2 t}{J}\sum_{j=1}^{J}\sum_{i=n-t+1}^{n-1}\mathbb{E}\left[\|\nabla w_{(i,j)}\|^2\right] + 2\gamma^2 t\sum_{i=n-t+1}^{n-1}\mathbb{E}\left[\|\nabla w_{(i,j_0)}\|^2\right]$$

$$\le 4\gamma^2 t^2 G^2.$$

$\square$

**Theorem A.2.** *Consider the sequence generated by the Delayed Algorithm, then*

$$\frac{1}{N}\sum_{n=0}^{N-1}\mathbb{E}[\|\nabla f(\overline{\nabla w_{(n)}})\|^2] \leq \frac{2}{\gamma N}(f(w_0) - f^*) + 4\gamma^2 t^2 G^2 L^2 + \frac{L}{J}\gamma\sigma^2.$$

*Proof.* By smoothness of $f$, we have

$$\mathbb{E}[f(\overline{\nabla w_{(n)}})] \leq \mathbb{E}[f(\overline{\nabla w_{(n-1)}}) + \langle \nabla f(\overline{\nabla w_{(n-1)}}), \overline{\nabla w_{(n)}} - \overline{\nabla w_{(n-1)}}\rangle + \frac{L}{2}\|\overline{\nabla w_{(n)}} - \overline{\nabla w_{(n-1)}}\|^2]$$

Note that

$$\overline{\nabla w_{(n)}} = \overline{\nabla w_{(n-1)}} - \frac{\gamma}{J}\sum_{j=1}^{J}\nabla w_{(n-1,j)}.$$

We have

$$\mathbb{E}[\|\overline{\nabla w_{(n)}} - \overline{\nabla w_{(n-1)}}\|^2]$$

$$= \gamma^2\mathbb{E}[\|\frac{1}{J}\sum_{j=1}^{J}\nabla w_{(n-1,j)}\|^2]$$

$$= \gamma^2\mathbb{E}[\|\frac{1}{J}\sum_{j=1}^{J}(\nabla w_{(n-1,j)} - \nabla f_j(w_{(n-1,j)}))\|^2] + \gamma^2\mathbb{E}[\|\frac{1}{J}\sum_{j=1}^{J}\nabla f_j(w_{(n-1,j)})\|^2]$$

$$= \frac{\gamma^2}{J^2}\sum_{j=1}^{J}\mathbb{E}[\|(\nabla w_{(n-1,j)} - \nabla f_j(w_{(n-1,j)})\|^2] + \gamma^2\mathbb{E}[\|\frac{1}{J}\sum_{j=1}^{J}\nabla f_j(w_{(n-1,j)})\|^2]$$

$$\leq \frac{\gamma^2\sigma^2}{J} + \gamma^2\mathbb{E}[\|\frac{1}{J}\sum_{j=1}^{J}\nabla f_j(w_{(n-1,j)})\|^2].$$

Moreover,

$$\mathbb{E}[\langle\nabla f(\overline{\nabla w_{(n-1)}}), \overline{\nabla w_{(n)}} - \overline{\nabla w_{(n-1)}}\rangle]$$

$$= -\gamma\mathbb{E}[\langle\nabla f(\overline{\nabla w_{(n-1)}}), \frac{1}{J}\sum_{j=1}^{J}\nabla w_{(n-1,j)}\rangle]$$

$$= -\gamma\mathbb{E}[\langle\nabla f(\overline{\nabla w_{(n-1)}}), \frac{1}{J}\sum_{j=1}^{J}\nabla f_j(w_{(n-1,j)})\rangle]$$

$$= -\frac{\gamma}{2}\mathbb{E}[\|\nabla f(\overline{\nabla w_{(n-1)}})\|^2] + \frac{\gamma}{2}\mathbb{E}[\|\nabla f(\overline{\nabla w_{(n-1)}}) - \frac{1}{J}\sum_{j=1}^{J}\nabla f_j(w_{(n-1,j)}\|^2] - \frac{\gamma}{2}\mathbb{E}[\|\frac{1}{J}\sum_{j=1}^{J}\nabla f_j(w_{(n-1,j)})\|^2]$$

Applying the lemma, we have

$$\mathbb{E}[\|\nabla f(\overline{\nabla w_{(n-1)}}) - \frac{1}{J}\sum_{j=1}^{J}\nabla f_j(w_{(n-1,j)}\|^2]$$

$$= \mathbb{E}[\|\frac{1}{J}\sum_{j=1}^{J}(\nabla f_j(\overline{\nabla w_{(n-1)}}) - \nabla f_j(w_{(n-1,j)}))\|^2]$$

$$\leq \frac{1}{J}\sum_{j=1}^{J}\mathbb{E}[\|(\nabla f_j(\overline{\nabla w_{(n-1)}}) - \nabla f_j(w_{(n-1,j)}))\|^2]$$

$$\leq \frac{L^2}{J}\sum_{j=1}^{J}\|\overline{\nabla w_{(n-1)}} - w_{(n-1,j)}\|^2$$

$$\leq 4\gamma^2 L^2 t^2 G^2.$$

Thus, combining the above inequalities together with $\gamma \leq \frac{1}{L}$ yields

$$
\mathbb{E}[f(\overline{\nabla w_{(n)}})] - \mathbb{E}[f(\overline{\nabla w_{(n-1)}})]
$$
$$
\leq \frac{\gamma L^2 - \gamma}{2}\mathbb{E}[\|\frac{1}{J}\sum_{j=1}^{J}\nabla f_j(w_{(n-1,j)})\|^2] - \frac{\gamma}{2}\mathbb{E}[\|\nabla f(\overline{\nabla w_{(n-1)}})\|^2] + 2\gamma^3 L^2 t^2 G^2 + \frac{L}{2J}\gamma^2\sigma^2
$$
$$
\leq -\frac{\gamma}{2}\mathbb{E}[\|\nabla f(\overline{\nabla w_{(n-1)}})\|^2] + 2\gamma^3 L^2 t^2 G^2 + \frac{L}{2J}\gamma^2\sigma^2.
$$

Rearrange the terms yields

$$
\mathbb{E}[\|\nabla f(\overline{\nabla w_{(n-1)}})\|^2] \leq \frac{2}{\gamma}(\mathbb{E}[f(\overline{\nabla w_{(n-1)}})] - \mathbb{E}[f(\overline{\nabla w_{(n)}})]) + 4\gamma^2 L^2 t^2 G^2 + \frac{L}{J}\gamma\sigma^2
$$

Telescoping from $n = 1, ..N$ yields

$$
\frac{1}{N}\sum_{n=1}^{N}\mathbb{E}[\|\nabla f(\overline{\nabla w_{(n-1)}})\|^2] \leq \frac{2}{\gamma}(\mathbb{E}[f(\overline{\nabla w_{(0)}})] - \mathbb{E}[f(\overline{\nabla w_{(n)}})]) + 4\gamma^2 L^2 t^2 G^2 + \frac{L}{J}\gamma\sigma^2
$$

$\square$

**Corollary A.2.1.** *When the number of iteration $N$ surpass the number of machines $J$, let the stepsize $\gamma = \frac{\sqrt{J}}{L\sqrt{N}}$, yields*

$$
\frac{1}{N}\sum_{n=0}^{N-1}\mathbb{E}[\|\nabla f(\overline{\nabla w_{(n)}})\|^2] \leq \frac{2L}{\sqrt{JN}}(f(w_0) - f^* + \sigma^2) + \frac{4t^2 G^2 J}{N}.
$$
$$
= O(\frac{1}{\sqrt{JN}}) + O(\frac{t^2 J}{N})
$$

When delayed steps is in a reasonable range ($t < O(N^{\frac{1}{4}}J^{-\frac{3}{4}})$), the first term dominates and delayed update demonstrates the same convergence speed as original SSGD $O(1/\sqrt{JN})$.

### A.2.2 PROOF OF THE CONVERGENCE OF TEMPORALLY SPARSE UPDATE ALGORITHM

---

**Temporally Sparse Update Algorithm**

1. Initialize $w_0$ and initialize each worker with $w_{0,j} = w_0$ for $j \in [1, J]$.

2. **For** iteration $n = 0, 1, \cdots$

    On each local worker $j$, sample a stochastic gradient $\nabla w_{(n-1,j)}$ and update:
    - If $n \pmod d \neq 0$, update with standard (local) SGD rule:
    $$
    w_{(n,j)} = w_{(n-1,j)} - \gamma\nabla w_{(n-1,j)}
    $$
    - Else:
    $$
    w_{(n,j)} = w_{(n-1,j)} - \gamma[\nabla w_{(n-1,j)} + \sum_{k=n-d}^{n-1}(\overline{\nabla w_{n-d+k}} - \nabla w_{(n-d+k,j)})] \quad (16)
    $$
    where
    $$
    \overline{\nabla w_{n-d+k}} = \frac{1}{J}\sum_{j=1}^{J}\nabla w_{(n-d+k,j)}
    $$

3. **Output** $\overline{\nabla w_{(n)}} = \frac{1}{J}\sum_{j=1}^{J}w_{(n,j)}$

---

Notably, the temporally sparse update formula can rewritten as

$$w_{(n,j)} = \underbrace{w_0 - \gamma \sum_{i=0}^{d\lfloor \frac{n}{d} \rfloor} \overline{\nabla w_i}}_{\text{invariant of } j} - \gamma \sum_{i=d\lfloor \frac{n}{d} \rfloor+1}^{n-1} \nabla w_{(i,j)}, \tag{TS}$$

where the first quantity is commonly shared among all the workers and the second term corresponds to local updates.

**Lemma A.3.** *Let* $(w_{(i,j)})$ *be the sequence generated on the $j$-th worker according to Temporally Sparse Update Algorithm, then for any* $j_0 \in [1, J]$

$$\mathbb{E}\left[\|\overline{\nabla w_{(n)}} - w_{(n,j_0)}\|^2\right] \leq 4\gamma^2 d^2 G^2$$

*Proof.* From the update of (TS), we have

$$\mathbb{E}\left[\|\overline{\nabla w_{(n)}} - w_{(n,j_0)}\|^2\right]$$

$$= \gamma^2 \mathbb{E}\left[\left\|\frac{1}{J}\sum_{j=1}^{J}\sum_{i=d\lfloor \frac{n}{d} \rfloor+1}^{n-1} \nabla w_{(i,j)} - \sum_{i=d\lfloor \frac{n}{d} \rfloor+1}^{n-1} \nabla w_{(i,j_0)}\right\|^2\right]$$

$$\leq 2\gamma^2 \mathbb{E}\left[\left\|\frac{1}{J}\sum_{j=1}^{J}\sum_{i=d\lfloor \frac{n}{d} \rfloor+1}^{n-1} \nabla w_{(i,j)}\right\|^2\right] + \mathbb{E}\left[\|\sum_{i=d\lfloor \frac{n}{d} \rfloor+1}^{n-1} \nabla w_{(i,j_0)}\|^2\right]$$

$$\leq \frac{2\gamma^2 d}{J}\sum_{j=1}^{J}\sum_{i=d\lfloor \frac{n}{d} \rfloor+1}^{n-1} \mathbb{E}\left[\|\nabla w_{(i,j)}\|^2\right] + 2\gamma^2 d \sum_{i=d\lfloor \frac{n}{d} \rfloor+1}^{n-1} \mathbb{E}\left[\|\nabla w_{(i,j_0)}\|^2\right]$$

$$\leq 4\gamma^2 d^2 G^2.$$

$\square$

**Theorem A.4.** *Consider the sequence generated by the Temporally Sparse Update Algorithm, then*

$$\frac{1}{N}\sum_{n=0}^{N-1} \mathbb{E}[\|\nabla f(\overline{\nabla w_{(n)}})\|^2] \leq \frac{2}{\gamma N}(f(w_0) - f^*) + 4\gamma^2 d^2 G^2 L^2 + \frac{L}{J}\gamma\sigma^2.$$

*Proof.* Based on Lemma A.3, the proof follows the exact same schema as the Delayed Update Algorithm, which we omit here. $\square$

**Corollary A.4.1.** *When the number of iteration $N$ surpass the number of machines $J$, let the stepsize* $\gamma = \frac{\sqrt{J}}{L\sqrt{N}}$*, yields*

$$\frac{1}{N}\sum_{n=0}^{N-1} \mathbb{E}[\|\nabla f(\overline{\nabla w_{(n)}})\|^2] \leq \frac{2L}{\sqrt{JN}}(f(w_0) - f^* + \sigma^2) + \frac{4d^2 G^2 J}{N}.$$

$$= O(\frac{1}{\sqrt{JN}}) + O(\frac{d^2 J}{N})$$

When temporal sparsity is in a reasonable range ($d < O(N^{\frac{1}{4}}J^{-\frac{3}{4}})$), the first term dominates and temporally sparse update demonstrates the same convergence speed as original SSGD $O(1/\sqrt{JN})$.

