# OpenReview forum: "Distributed Training Across the World"
_ICLR.cc/2020/Conference — Reject_

### Official Review · AnonReviewer3 · 2019-10-24
**Official Blind Review #3**

**Rating:** 3

**Review:**

The paper presents an approach - Delayed and Temporally Sparse Update (DTS) - to do distributed training on nodes that are very distant from each other in terms of latency. The approach relies on delayed updates to tolerate latencies and temporally sparse updates to reduce traffic. The approach is implemented in a synchronous stochastic gradient decent (SGD) scheme.

The paper's approach with delayed updates, i.e., delaying gradient updates to other nodes, sends the updates in a later iteration. In this way, very long latencies can be hidden (covered by computation) since the gradient updates can be postponed to an arbitrary iteration (barrier synchronization) in the future.

Unfortunately, delaying the updates opens up for the same problems as asynchronous SGD (ASGD), i.e., slower (none) convergence or staleness problems. These problems are coped with using a compensation factor, that adjusts the momentum and weights accordingly.

A critical aspect of ASGD is to provide convergence guarantees, and DTS is similar in that aspect. However, the paper does not provide any convergence analysis or convergence guarantees.

Using only one dataset / network to evaluate the approach is too little. In order to show the generality, more benchmarks need to be evaluated.

In general, I like the paper. It is well written and easy to read. However, the novelty and contribution is relatively low. A lot of work has been done in the HPC and distributed systems communities over decades on how to tolerate latencies, trade-offs between communication and computation, proper synchronization points, etc. The ideas presented in this paper are well-known and extensively applied in those communities. None of that work is referensed or acknowledged in this paper.

The term "scalability" is used as an evaluation metric, but never clearly defined in the paper.


**Experience Assessment:**

I have published one or two papers in this area.

**Review Assessment: Checking Correctness Of Derivations And Theory:**

I assessed the sensibility of the derivations and theory.

**Review Assessment: Checking Correctness Of Experiments:**

I assessed the sensibility of the experiments.

**Review Assessment: Thoroughness In Paper Reading:**

I read the paper at least twice and used my best judgement in assessing the paper.

---

> ### Author Response · Authors · 2019-11-14
> **Response to reviewer #3**
>
> Thanks for your comments!  For the theoretical parts, we attach proof of convergence and the connection between our algorithm and variance reduction (please see Appendix updated PDF). We show that our proposed delayed update and temporally sparse update have the same convergence rate as original synchronous distributed SGD.
>
> >>>  none convergence /  convergence analysis / compared with SSGD
>
> Both R2 and you have raised this concern. To address it, we have added a convergence analysis of our algorithm. Under the non-convex smooth optimization setting, we show that our algorithms enjoy the same convergence rate as SSGD. Moreover, we make a new connection of our methods to the well-known variance reduction technique [4]. We believe that this connection provides intuition and insights on the effectiveness of our algorithm. All the details are included in appendix.
>
> The convergence rate of our method is given by
>
> $$ O(\frac{1}{\sqrt{NJ}}) + O{\frac{t^2J}{N}} $$
>
> where $t$ is the delay parameter, $N$ is the number of iterations, $J$ is the number of workers. When $t$ is reasonable ( $t < O(N^{\frac{1}{4}} J^{-\frac{3}{4}})$ ), the first term dominates and the convergence rate is identical to the convergence rate of SSGD.
>
> >>> delaying the updates opens up for the same problems as asynchronous SGD (ASGD)
>
> The biggest known problem for ASGD is the stale gradients, though it is also where the scalability benefits from. The main challenge is to reduce the noise brought by staleness. There are studies on (1) dampening the stale gradients [1] (2) correct the gradients [2] (3) limit the maximum staleness [3]. In order to deal with the stale gradient, our work incorporates the well-known variance reduction technique, which naturally reduces the noise. Our theoretical analysis justifies the soundness of the proposed strategy.
>
> Moreover, the latency grows, ASGD suffers from performance degrade because of increasing staleness of gradients. On ImageNet,
>  The strength of our method is to allow high scalability with negligible loss on performance.
>
> >>> Using only one dataset/network to evaluate the approach is too little
>
> On one hand, ImageNet is a large scale dataset and ResNet is a widely adapted models in both industry and research. We believe the improvements on ResNet / ImageNet will consistently translate into other models and tasks.
>
> On the other hand, we have performed additional experiments on NLP tasks. We experiment DTS on 2 layer LSTM on the Penn Treebank corpus (PTB) dataset. The results are attached below. DTS well-preserves the accuracy up to 20 delay steps and temporal sparsity of 1/20.
>
>
>                      Perplexity |                                                  Perplexity |
> Original          72.30      |   Original                                   72.30       |
> Delay(t)=4      72.28      |   Temporal Sparse (p)=4         72.27       |
> Delay(t)=10    72.29      |   Temporal Sparse (p)=10       72.31       |
> Delay(t)=20    72.27      |   Temporal Sparse (p)=20       72.28       |
>
> >>> Definition of scalability
>
> The idea case in distributed training is that N machines bring N times speedup. However, this is usually not achievable in practice because of communication costs (latency and bandwidth). If a system can achieve M times speedup on N machines, the scalability is M / N, which demonstrates how scalable the system is. We will clarify this point in the revision.
>
> >>> Related work / Novelty
>
> We agree with the reviewer that sparse communication has been studied previously and our temporal sparsity variant is similar to them as a strategy to amortize latency. We will add more related works including [5, 6, 7]. Please kindly advise if we miss any. However, to tolerate the latency, we have proposed a variant with delayed update, which is completely novel and new, according to the best of our knowledge.
>
> Existing methods with sparse communication do not scale well when latency grows since they only amortize it. Conventional ASGD based approaches, though scales well with growth of latency, suffer from performance drop because of increasing staleness. DTS is the first work to both maintain good scalability (0.72) and achieve promising results (no drop on accuracy) on modern models (ResNet-50) and large scale dataset (ImageNet).
>
> [1] Communication compression for decentralized training.
> [2] Staleness-aware async-sgd for distributed deep learning.
> [3] More effective distributed ml via a stale synchronous parallel parameter server.
> [4] Accelerating stochastic gradient descent using predictive variance reduction.
> [5] Communication-efficient learning of deep networks from decentralized data."
> [6] Parallel restarted SGD with faster convergence and less communication: Demystifying why model averaging works for deep learning.
> [7] Deep learning with elastic averaging SGD.

---

### Official Review · AnonReviewer2 · 2019-10-24
**Official Blind Review #2**

**Rating:** 3

**Review:**

This paper tackles the issue of scalability in distributed SGD over a high-latency network.
Along with experiments, this paper contributes two ideas: delayed updates and temporally sparse updates.

- strengths:

· Clear-looking overall presentation.
  Good efforts to explain the network problems (latency, congestion) and how they are tackled by delayed updates and temporally sparse updates.

- weaknesses:

· While the overall presentation looks clean, the paper does not talk about the setting (one parameter server and many workers, only workers, etc).
  This is arguably basic information, and the readers is left to understand by themselves that the setting consists in many workers without a parameter server.

· There is no theoretical analysis of the soundness of the proposed algorithm.
  For instance in ASGD (that the authors cite), stale gradients are dampened before being used, which in turn is used to guarantee convergence.
  In the proposed algorithm, there is no dampening nor apparent limit of the maximum value of t; such a difference with prior art should entail a serious (theoretical) analysis.

· Finally, using SSGD for comparison is not very "fair", as communication-efficient algorithms have already been published for quite some time [1, 2, and follow-ups (e.g. searching for "local sgd")].
  At the very least a comparison with ASGD (cited) is necessary, as in a realistic setting latency is indeed a problem but arguably not bandwidth
  (plus, orthogonal gradient compression techniques do exist, e.g. as in "Federated Learning: Strategies for Improving Communication Efficiency").

questions to the authors:

- Can you clarify the setting?

- Can you give at least an intuition why accepting stale gradient is correct (i.e. does not impede convergence)?
  There is no theoretical limit on the value of t; can workers take any arbitrary old gradient?
  So when the training is close to convergence (if it reaches it), i.e., when the norm of the gradients are close to 0,  the algorithm could then use very old gradients (which norms could be orders of magnitude larger) to "correct the mismatch" caused by the local training; is this correct?
  To solve that issue, ASGD introduces a simple dampening mechanism (which is necessary in the convergence proof), which your algorithm does not have.


The related work section focuses on gradient compression techniques (which tackle low bandwidth, not latency) and asynchronous SGD (which is more prone to congestion, with a single parameter server),
but seems to overlook that sparse communication techniques already exist (this fact should at least be mentioned).

The idea of sparse communication for SGD exists since at least 2012 [1, Algorithm 3].
A first mention of the use of such techniques for "communication efficiency" dates from (at least) 2015 [2].

[1] Ofer Dekel, Ran Gilad-Bachrach, Ohad Shamir, and Lin Xiao.
    Optimal distributed online prediction using mini-batches.
    J. Mach. Learn. Res., 13(1):165–202, January 2012.

[2] Sixin Zhang, Anna E. Choromanska, Yann LeCun.
    Deep learning with Elastic Averaging SGD.
    NeurIPS, 2015.


I do not see a clear novelty, nor a proof (or even intuitions) that the proposed algorithm is theoretically sound.
The comparison with SSGD is arguably unfair, since SSGD is arguably not at all the state-of-the-art in the proposed setting (hence the claimed speedup of x90 can be very misleading).

**Experience Assessment:**

I have published in this field for several years.

**Review Assessment: Checking Correctness Of Derivations And Theory:**

I assessed the sensibility of the derivations and theory.

**Review Assessment: Checking Correctness Of Experiments:**

I assessed the sensibility of the experiments.

**Review Assessment: Thoroughness In Paper Reading:**

I read the paper at least twice and used my best judgement in assessing the paper.

---

> ### Author Response · Authors · 2019-11-14
> **Response to reviewer #2**
>
> Thanks for your detailed and constructive comments! For the theoretical parts, we attach proof of convergence and the connection between our algorithm and variance reduction (please see Appendix updated PDF). We show that our proposed delayed update and temporally sparse update have the same convergence rate as original synchronous distributed SGD.
>
> >>> Experimental settings
>
> We are using homogeneous infrastructure for training: All servers are communicated through allreduce and there is no central parameter server. We will make it clear in revision.
>
> >>> Theoretical soundness
>
> Both R3 and you have raised this concern. To address it, we have added a convergence analysis of our algorithm. Under the non-convex smooth optimization setting, we show that our algorithms enjoy the same convergence rate as SSGD. Moreover, we make a new connection of our methods to the well-known variance reduction technique [4]. We believe that this connection provides intuition and insights on the effectiveness of our algorithm. All the details are included in appendix.
>
> >>> Limitation on value of t / Arbitrarily old gradients when close to convergence
>
> The convergence rate of our method is given by
>
> $$ O(\frac{1}{\sqrt{NJ}}) + O{\frac{t^2J}{N}} $$ ,
>
> where $t$ is the delay parameter, $N$ is the number of iterations, $J$ is the number of workers. When $t$ is in a reasonable range ( $t < O(N^{\frac{1}{4}} J^{-\frac{3}{4}})$ ), the first term dominates and the convergence rate is identical to the convergence rate of SSGD.
>
> When $t=0$, there is no staleness, the update is identical to SSGD.
> When $t=N$, the algorithm becomes identical as local SGD and does not converge.
>
> As shown in the convergence analysis, $t$ has to be a reasonable value and workers cannot take any arbitrary old gradient. However, $t$ can be set to larger when training close to convergence (the first term in convergence rate dominates). Your observation inspires us to design a warm-up strategy: $t$ is initially set to a small value and grows during optimization. With this technique, we can double the maximum staleness on CIFAR (no loss on accuracy).
>
> >>> Intuitions / Why no dampening / Difference between ASGD
>
> The biggest known problem for ASGD is the stale gradients, though it is also where the scalability benefits from. The main challenge is to reduce the noise brought by staleness. There are studies on (1) dampening the stale gradients [1] (2) correct the gradients [2] (3) limit the maximum staleness [3]. In order to deal with the stale gradient, our work incorporates the well-known variance reduction technique, which naturally reduces the noise. Our theoretical analysis justifies the soundness of the proposed strategy.
>
> One thing worth noting is that though the scalability of ASGD variants does not vary much with the growth of latency, the accuracy will drop because increasing staleness, which is not we want. That is why we mainly compare with SSGD in our initial submission. The strength of our method is to allow high scalability with a negligible loss of accuracy.
>
> We agree with the reviewer that we should have compared to ASGD and it is an omission from our part. We now have added comparisons to vanilla ASGD and ECD-PSGD (an improved version of ASGD) in Fig 1. When training across the world, the performance of ECD-PSGD drops quickly where DTS has no drop on accuracy while maintaining good scalability.
>
> >>> Sparse Communication
>
> Distributed training has been focusing on training inside a cluster where latency is also promised to be low. Most previous works consider more about bandwidth rather than latency. Though conventional approaches like sparse communication can be also applied to amortize latency, it cannot fully cover the communication by computation especially when the latency is high (e.g., federated learning).
>
> We agree with the reviewer that sparse communication has been studied previously and our temporal sparsity variant is similar to them. However, the variant on the delayed update is completely novel, according to the best of our knowledge.
>
> More importantly, as shown in Fig.4, delayed update is more effective than sparse communication (temporal sparse update) when against latency. It is worth noting that delayed update is NOT a sparse communication: information are sent received at every iteration just as SSGD. The main difference is that the received information is processed in a delayed manner. While sparse communication can only amortize the latency, our delayed technique can fully cover the cost of communication by computation.
>
> [1] Tang, Hanlin, et al. "Communication compression for decentralized training."
> [2] Zhang, Wei, et al. "Staleness-aware async-sgd for distributed deep learning."
> [3] Ho, Qirong, et al. "More effective distributed ml via a stale synchronous parallel parameter server."
> [4] Johnson, Rie, and Tong Zhang. "Accelerating stochastic gradient descent using predictive variance reduction."

---

> > ### Comment · AnonReviewer2 · 2019-11-15
> > **Thank you for the detailed answer**
> >
> > Dear authors, I acknowledge your detailed answer and believe this paper do have some merits. Glad that the comment on t inspired that warm up strategy. The theoretical analysis answers some of my concerns, but I remain doubtful that it adresses the question of gradient coherence, this is a concern I have not with your paper alone but with a few other on asynchronous SGD: the hypothesis that current and stale gradients will somehow point to the same half-space and decrease the loss needs much more exploration than what the community did so far (assuming it holds almost for free). I would highly suggest you explore this in any future iteration of this (relevant) work.

---

> > > ### Author Response · Authors · 2019-11-15
> > > **Re: Thank you for the detailed answer**
> > >
> > > Thank you for your comment! We would try to address your concern on gradient coherence.
> > >
> > > First, let us consider the standard SGD algorithm (on a single machine). When we sample a mini-batch, there is no guarantee that the specific stochastic gradient on this specific mini-batch will point to the optimum. What we could guarantee is in expectation they are pointing to the optimum.
> > >
> > > Now let us consider the delay update with delay t. It is true that the iterates on each machine are different  (i.e. not synchronized), but our update strategy guarantees that they are not far from each other. This is characterized by Lemma A.1 in the appendix, showing that the difference of local iterates are bounded by a constant proportional to t. This implies that the stale gradient does not deviate too much from the current gradients in expectation when t is reasonable (because of Lipschitz smoothness).
> > >
> > > We hope this clarifies your concern. Finally, our method is 1) theoretically sound; 2) effective on tolerating high latency and achieving high scalability, which we believe has a lot of potential in large-scaled applications. We will appreciate if you could reconsider the score based on this contribution. Thank you!

---

### Official Review · AnonReviewer1 · 2019-10-29
**Official Blind Review #1**

**Rating:** 6

**Review:**

Authors provide a technique to compensate for error introduced by stale or infrequent synchronization updates in distributed deep learning.

Their solution is to use local gradient information to update the model, and once delayed gradient information from other workers arrives, the use it to provide a correction which would give an equivalent result to "no delay" synchronization in the case of linear model training.

The three approaches are:
1. Delayed update -- use local gradients for immediate update, apply correction when stale averaged update arrives.
2. Sparse update -- only update once every p iterations, the averaged update includes p steps
3. Combined -- 1. and 2. combined

Authors evaluate on ImageNet, showing some improvement, and promise to release their implementation for PyTorch/Horovod. Their technique, combined with a reference implementation in a popular framework stands a good chance of having impact. Given the increase in cloud training workloads, even a small improvement in this setting is significant.

Comments:
- "scalability" is never defined. I would recommend defining it, or referencing a paper which defines it. I assume it refers to training throughput divided by ideal training throughput.
- Evaluation is one on resnet-50. Because it's mostly convolutions, such network has high compute/network ratio and is not frequently bottlenecked by network. A more convincing experiment would rely on lower computation intensity architecture such as Transformer/BERT training.
- Section 1 states that without their technique, they expect SGD to exhibit 0.008 scalability for 100 servers, compared to 0.72 for their method. However, the number 0.72 was not supported by data, their largest experiment used 16 servers.



**Experience Assessment:**

I have published one or two papers in this area.

**Review Assessment: Checking Correctness Of Derivations And Theory:**

I assessed the sensibility of the derivations and theory.

**Review Assessment: Checking Correctness Of Experiments:**

I assessed the sensibility of the experiments.

**Review Assessment: Thoroughness In Paper Reading:**

I read the paper at least twice and used my best judgement in assessing the paper.

---

> ### Author Response · Authors · 2019-11-14
> **Response to reviewer #1**
>
> Thanks for your reviewer. Please see point-to-point response below:
>
> >>> The definition of scalability
>
> The ideal case in distributed training is that N machines bring N times speedup. However, this is usually not achievable in practice because of communication costs (latency and bandwidth). If a system can achieve M times speedup on N machines, the scalability is M / N, which demonstrates how scalable the system is. We will make this point clear in the revision.
>
> >>> Evaluation only for ResNet-50 on ImageNet
>
> We agree that ResNet-50 is a more compute-intensive model. However, as shown in  Fig1, 4.a, and 4.b, even for models with high compute/network ratio, the training efficiency of conventional approaches are still seriously affected by latency. It could be only worse when dealing with communication-intensive models.
>
> We have added more experiments on NLP tasks as suggested by the reviewer. We experimented DTS on 2 layer LSTM on the Penn Treebank corpus (PTB) dataset. The results are attached below. DTS well-preserves the accuracy up to 20 delay steps and temporal sparsity of 1/20.
>
>
>                      Perplexity |                                                  Perplexity |
> Original          72.30      |   Original                                   72.30       |
> Delay(t)=4      72.28      |   Temporal Sparse (p)=4         72.27       |
> Delay(t)=10    72.29      |   Temporal Sparse (p)=10       72.31       |
> Delay(t)=20    72.27      |   Temporal Sparse (p)=20       72.28       |
>
> >>> Experiment settings
>
> Apology for the confusion. We experiment on 16 eight-card GPU servers and obtain scalability of 0.008. We intend to emphasize how poor the scalability of SSGD is, that even with 100 workers (assume the same scalability 0.008), it is still slower than a single machine. We will clarify this point.
>
> >>>  Further comments
>
> We have added convergence analysis of our algorithm in the appendix. This justifies the theoretical soundness of our algorithm. Moreover, we bridge a new connection between our delayed variant and the well-known variance reduction technique in convex optimization, we believe this provides more intuition and insight on the empirical effectiveness of our method.

---

### Public Comment · ~Boris_Ginsburg1 · 2019-09-29
**Very interesting work!**

Very interesting work!
A few questions:
1. Does this method require to remember t(delay) previous gradients to compute compensation error?
2. It looks like there is typo in  equations (1),(2)...: it should be "w - lambda*grad" instead of "w + lambda*grad"?
3. Have you tried  to apply this method to other tasks (NLP, speech...)?

---

> ### Author Response · Authors · 2019-11-01
> **Thanks for your interest!**
>
> 1. Yes. It is required to store previous gradients for calculating error compensation. Notably it just stores locally and does not bring any extra cost to the communication.
> 2. We use “w’ = w + lambda * grad” as the update formula. We will rewrite the equations to be consistent with the mainstream. Apologize for the confusion.
> 3. Yes, it works for NLP task. We have experimented DTS on 2 layer LSTM on the Penn Treebank corpus (PTB) dataset. The results are attached below. DTS well-preserves the accuracy up to 20 delay steps and temporal sparsity of 1/20.
>
>
>                      Perplexity |                                                  Perplexity |
> Original          72.30      |   Original                                   72.30       |
> Delay(t)=4      72.28      |   Temporal Sparse (p)=4         72.27       |
> Delay(t)=10    72.29      |   Temporal Sparse (p)=10       72.31       |
> Delay(t)=20    72.27      |   Temporal Sparse (p)=20       72.28       |

---

### Author Response · Authors · 2019-11-15
**Our General Response**

We sincerely thank all reviewers for their comments. All reviewers agree that the paper is clearly written and has certain contributions to against latency. R2 & R3 mainly concern about the convergence guarantees, which we have justified through proof in the appendix (guarantee to converge and no slower than SGD). For unclear parts in writing, we have revised accordingly in the updated version.

We have added experiments on NLP tasks and DTS well-preserves the accuracy while tolerating a high latency.  Some experiments (e.g., transformer and BERT) takes a longer time than the rebuttal period. We will update these results later.

Our code is available on : https://drive.google.com/drive/folders/1XdneMoRPNooN3-dj6yFwFcQRQO6OhLB1?usp=sharing

If there are any additional comments/suggestions on paper/code,  please let us know!

---

### Decision · Program_Chairs · 2019-12-19

**Decision:**

Reject

**Comment:**

The paper introduces a distributed algorithm for training deep nets in clusters with high-latency (i.e. very remote) nodes. While the motivation and clarity are the strengths of the paper, the reviewers have some concerns regarding novelty and insufficient theoretical analysis.